# Modulation transfer functions for audiovisual speech

**Nicolai F. Pedersen**[1], **Torsten Dau**[1], **Lars Kai Hansen**[2], **Jens Hjortkjær**[1,3]*

**1** Hearing Systems, Department of Health Technology, Technical University of Denmark, Kgs. Lyngby, Denmark, **2** Department of Applied Mathematics and Computer Science, Technical University of Denmark, Kgs. Lyngby, Denmark, **3** Danish Research Centre for Magnetic Resonance, Centre for Functional and Diagnostic Imaging and Research, Copenhagen University Hospital Amager and Hvidovre, Copenhagen, Denmark

* jhjort@dtu.dk

**Data Availability Statement:** The code and linked data underlying the results presented in the study is available from https://github.com/NicolaiP/cca_mtfs.

## Abstract

Temporal synchrony between facial motion and acoustic modulations is a hallmark feature of audiovisual speech. The moving face and mouth during natural speech is known to be correlated with low-frequency acoustic envelope fluctuations (below 10 Hz), but the precise rates at which envelope information is synchronized with motion in different parts of the face are less clear. Here, we used regularized canonical correlation analysis (rCCA) to learn speech envelope filters whose outputs correlate with motion in different parts of the speakers face. We leveraged recent advances in video-based 3D facial landmark estimation allowing us to examine statistical envelope-face correlations across a large number of speakers (~4000). Specifically, rCCA was used to learn modulation transfer functions (MTFs) for the speech envelope that significantly predict correlation with facial motion across different speakers. The AV analysis revealed bandpass speech envelope filters at distinct temporal scales. A first set of MTFs showed peaks around 3-4 Hz and were correlated with mouth movements. A second set of MTFs captured envelope fluctuations in the 1-2 Hz range correlated with more global face and head motion. These two distinctive time-scales emerged only as a property of natural AV speech statistics across many speakers. A similar analysis of fewer speakers performing a controlled speech task highlighted only the well-known temporal modulations around 4 Hz correlated with orofacial motion. The different bandpass ranges of AV correlation align notably with the average rates at which syllables (3-4 Hz) and phrases (1-2 Hz) are produced in natural speech. Whereas periodicities at the syllable rate are evident in the envelope spectrum of the speech signal itself, slower 1-2 Hz regularities thus only become prominent when considering crossmodal signal statistics. This may indicate a motor origin of temporal regularities at the timescales of syllables and phrases in natural speech.

## Author summary

Natural speech signals are dominated by slow fluctuations (<10 Hz) in the acoustic speech envelope. A peak in modulation energy around 3–4 Hz corresponds to the average rate at

**Funding:** TD was supported by the Novo Nordisk Foundation synergy grant NNF17OC0027872 (UHeal). LKH was supported by the Danish Pioneer Centre for AI, DNRF grant number P1. TD was supported by the Oticon Centre of Excellence for Hearing and Speech Sciences (CHeSS) grant from William Demant Foundation. The funders had no role in study design, data collection and analysis, decision to publish, or preparation of the manuscript.

**Competing interests:** The authors have declared that no competing interests exist.

which syllables are produced in natural speech, but speech carries temporal information at multiple timescales. Here, we show that audiovisual speech statistics derived from natural speech across many speakers reveal different and distinct timescales of envelope fluctuations correlated with different kinematic components of the speaker's face. Using regularized canonical correlation analysis, we analyzed a comprehensive natural speech video data set to derive modulation transfer functions for the speech envelope conditioned on correlations with facial motion. Distinct timescales of audiovisual correlation emerged: (i) speech envelope fluctuations around 3–4 Hz correlated with mouth openings, as expected, and (ii) slower 1–2 Hz envelope fluctuations correlated with more global facial motion. These different envelope frequency regions align notably with the timescales of syllables and phrases in natural speech and may point to a motor origin of these privileged rates.

## Introduction

Seeing a person's face is known to influence auditory speech perception [1] and can improve speech intelligibility in noisy environments [2]. Visual information can also inform automatic speech recognition [3] or speech separation systems [4]. Audiovisual speech perception is thought to hinge on temporal correspondences between the auditory and visual signals received by the perceiver. Both amplitude envelope fluctuations in the acoustic speech signal and the motion of orofacial articulators during speech production are dominated by slow 'rhythms' predominant in the 1–8 Hz range [5–7]. However, the details of how speech modulations at different rates within this range correlate with visible movement in different parts of the talker's face or head are still not fully understood.

Orofacial movements during speech production display relatively slow quasi-regular kinematics. Studies measuring jaw, lip, or tongue movements during speech have reported regular motion patterns predominantly below 8 Hz [5, 8–11]. Ohala (1975), for example, reported histograms of intervals between jaw openings measured during running speech, showing a peak frequency in the 3–6 Hz range [12]. This corresponds to the average rate at which syllables are produced in natural speech, although variation exists across languages and speakers [13–15]. The natural syllable production rate has also been argued to determine the shape of the modulation spectrum of natural speech signals [16], consistently showing a peak frequency around 3–4 Hz across different languages and speech corpora [7, 15, 17].

However, the co-existence of slow periodicities in face movements and in the produced speech signal does not by itself specify the details of how they are related. It also does not reveal which dynamic visual cues are available in audiovisual speech perception or decodable from video inputs of a speaker's face. Some periodic movements occurring during speech may not be related to the production of sound or necessarily correlated with any acoustic events (e.g. blinking). Conversely, natural speech sounds contain amplitude modulations that may not be directly related to any visible movement available to the perceiver (such as speech modulations produced predominantly by phonatory activity). Although the two domains share a temporal axis, the temporal characteristics of the relation between visible motion and speech acoustics remain to be specified.

A number of previous studies have examined correlations between orofacial movements and different features of the acoustic speech signal [5, 6, 18–21]. Most work has considered temporal envelope representations extracted by low-pass filtering the speech audio waveform. Chandrasekaran et al. (2009) reported a correlation between speech envelopes and the area of

mouth openings extracted from speech videos [6]. To extract the envelope, the speech signal was first filtered in the audio frequency domain, Hilbert transformed, and down-sampled to 25 Hz, but the envelope was not decomposed further. To examine the relation between the mouth area and the speech envelope as a function of temporal modulation frequency, the spectral coherence between the audio and video signal features was examined. This suggested that mouth openings and speech envelopes both contain temporal modulations in the 2–6 Hz range. Alexandrou et al. (2016) reported a similar range of spectral coherence between speech envelopes and electromyographic lip and tongue recordings [22]. Coherence analyses of this type demonstrate that auditory and visual signals display some degree of periodicity in the same spectral range. However, spectral coherence does not extract potential different sources of co-variance in the spectral range where coherence is observed. This requires a decomposition of the covariance structure in the envelope domain.

The majority of studies have focused on the correlation between speech acoustics and movements of the mouth. However, other parts of the face or body move as well during natural speech [23]. Some of these may be coupled with orofacial articulators in speech motor control. Other gestures performed during naturalistic speech may not be directly involved in sound production but may nonetheless be consistently correlated with sound features. Rhythmic head nodding or eyebrow movements during speech, for instance, have been associated with speech prosody [24–28]. Head or body movements may thus also correlate with variations in acoustic features [19, 20, 29, 30] but presumably at slower rates given the kinematics of head or body motion [31]. More generally, it remains unclear how different parts of the talking face and head may be correlated with different rates of acoustic variation in the speech signal during natural speech.

This question is complicated by the fact that different moving parts of the face are themselves mutually correlated during natural speech. Individual articulators do not move independently but are synergistically coordinated via common neuromuscular control [32] or biomechanical coupling [33]. For example, movements of the hyoid, jaw, and tongue display a unique and rate-specific degree of coupling during speech, and the coupling is distinct from other behaviors such as chewing [11, 34–36]. Since different parts of the speech motor system are coordinated, it is necessary to consider how different parts of the face form groups with common kinematics. Data-driven dimensionality reduction techniques have been used to analyze facial motion data recorded during speech production in order to identify spatial components that follow shared motion patterns [19, 37–40]. Lucero & Munhall (2008) used QR factorization to identify groups of linearly dependent facial markers, revealing a set of *kinematic eigenregions* in the speaking face [40]. Consistently across two talkers, such eigenregions were identified for the lower and upper parts of the mouth and each of the mouth corners. Regions in other non-oral parts of the face were also identified, such as the left and right eyebrows and the two eyes [40]. Such data-driven analyses of facial markers may capture the spatial *degrees of freedom* or dimensionality of facial kinematics during speech production, but may also identify spatial components that are not necessarily related to the acoustic speech signal.

In the current study, we present an AV analysis approach based on *canonical correlation analysis* (CCA) that linearly transforms *both* visual and audio signals to capture the correlational structure between them. This approach simultaneously segments facial landmarks (as in previous work) while filtering the speech audio signal in the envelope domain. We adopt an idea originally proposed for the analysis of electrophysiological responses to speech [41] that uses CCA to learn modulation transfer functions (MTFs) in the audio envelope domain. De Cheveigné et al. (2018) applied a multichannel FIR filter bank to speech envelopes as input to the CCA (the second input being EEG brain signals) [41]. Each component of the CCA then

linearly recombines the envelope subbands to find a filtered audio envelope that maximizes the correlation with the second input. With an appropriate choice of filters, the filter bank constitutes a *filter basis* and CCA learns optimal coefficients on that basis [41]. Here, we adapt this idea to learn envelope filters that correlate with visual motion in different regions of the speaker's face. Specifically, CCA simultaneously learns a set of envelope filters and a corresponding set of kinematic eigenregions of the face. The MTFs of the envelope filters learned by CCA can then be used to characterize the range of temporal modulation frequencies that correlate with different kinematic regions of the face.

MTFs have traditionally been used to characterize how an acoustic transmission channel, such as a room, attenuates or enhances certain modulation frequencies in the input sound signal [42]. MTFs have also been used to characterize the sensitivity to amplitude modulations in auditory perception [43–45] or physiology [46, 47]. In the context of AV speech analysis, we adapt the MTF concept to characterize the range of envelope frequencies in the speech signal that are correlated with visual motion. Similar to MTFs in auditory physiology or perception, we speculated that the relation between the acoustic speech envelope and the visual face might have a band-pass character, i.e. that narrower ranges of speech modulation frequencies might be related to visible motion in different parts of the face. In contrast to its application in room acoustics or perception, the MTFs of AV speech do not map the acoustic speech signal directly to the visual signal, but instead transform both signals to a latent representation learned by CCA. This is motivated by the fact that the visual signal is not directly caused by the acoustic signal, or vice versa. Instead, the audio and video signals are both related to the underlying speech production system [48] and its neuromuscular control [32, 49].

Here, we analyzed an extensive video dataset of natural speech using CCA. Our primary analysis was based on the LRS3 (lip-reading sentences) dataset consisting of single-talker video recordings collected *in the wild* (videos from TED and TEDx talks, [50]). We exploited novel deep learning techniques to estimate 3D facial landmarks directly from 2D videos of the speakers. In contrast to previous work based on manual motion tracking, the estimation of face points from video enabled us to model the statistics of facial kinematics and their relation to speech envelope variations across a large number of speakers (>4000). Specifically, we used regularized CCA (rCCA) to identify face-envelope correlations that generalize across speakers, i.e. patterns of head and face movement that are consistently correlated with speech modulations across a large number of speakers in the LRS3 dataset. We also compared the results to more well-controlled speech recordings (the GRID dataset, [51]) used in a number of previous AV speech studies.

## Materials and methods

### Data

**LRS3 dataset.**    The main analysis was conducted on the LRS3 dataset [50], containing *in the wild* videos of single speakers extracted from TED and TEDx talks in English. The predefined `trainval` training dataset consisting of 32,000 videos or approximately 30 hours of video data was used. The dataset is composed of video clips from 4,004 different speakers. The videos were recorded with a frame-rate of 25 fps, an audio sample rate at 16 kHz, and the clips vary from one to six seconds in duration. Videos were excluded if the face landmarks could not be estimated, leaving a total of 30,934 videos corresponding to approximately 29.5 hours of video data.

**GRID dataset.**    For comparison, the analysis was also performed on the GRID dataset [51], used in a number of previous AV speech studies (e.g. [6]). In contrast to LRS3, the GRID data consists of data from fewer speakers performing a controlled speech task. The data

consists of audio and video recordings of 34 native English speakers, each reading 1,000 predefined matrix sentences. Each sentence consists of six monosyllabic words: command, color, preposition, letter, digit, and adverb, e.g., *"place green by D 2 now"* out of a total vocabulary of 51 words. The speaker is situated in front of a neutral background and facing the camera. All videos have a duration of 3 seconds and are recorded with a video frame-rate of 25 frames per second (fps) and an audio sample rate of 50 kHz. [51]. Videos for one of the speakers (speaker 21) were not available. From the 33,000 available videos, a total of 32,731 videos were included in the analysis, corresponding to approximately 27 hours of video data.

## Feature extraction

**Audio envelope extraction.** We estimated an envelope representation of the speech audio signals (see Fig 1). First, the audio files were resampled to 16.000 Hz and converted to mono. The speech waveform signals were passed through a gammatone filterbank [52] consisting of 31 filters spaced from 80 to 8000 Hz. The envelope was then computed in each gammatone subband via the Hilbert transform. Next, the envelopes in each subband were passed through a modulation filterbank comprising a set of 25 equally spaced first-order Butterworth bandpass filters with a bandwidth of 0.75 Hz and a spacing of 0.5 Hz. Each envelope subband was then averaged across the gammatone filters and resampled to 25 Hz to match the video framerate, and finally normalized to have zero mean and unit variance per video.

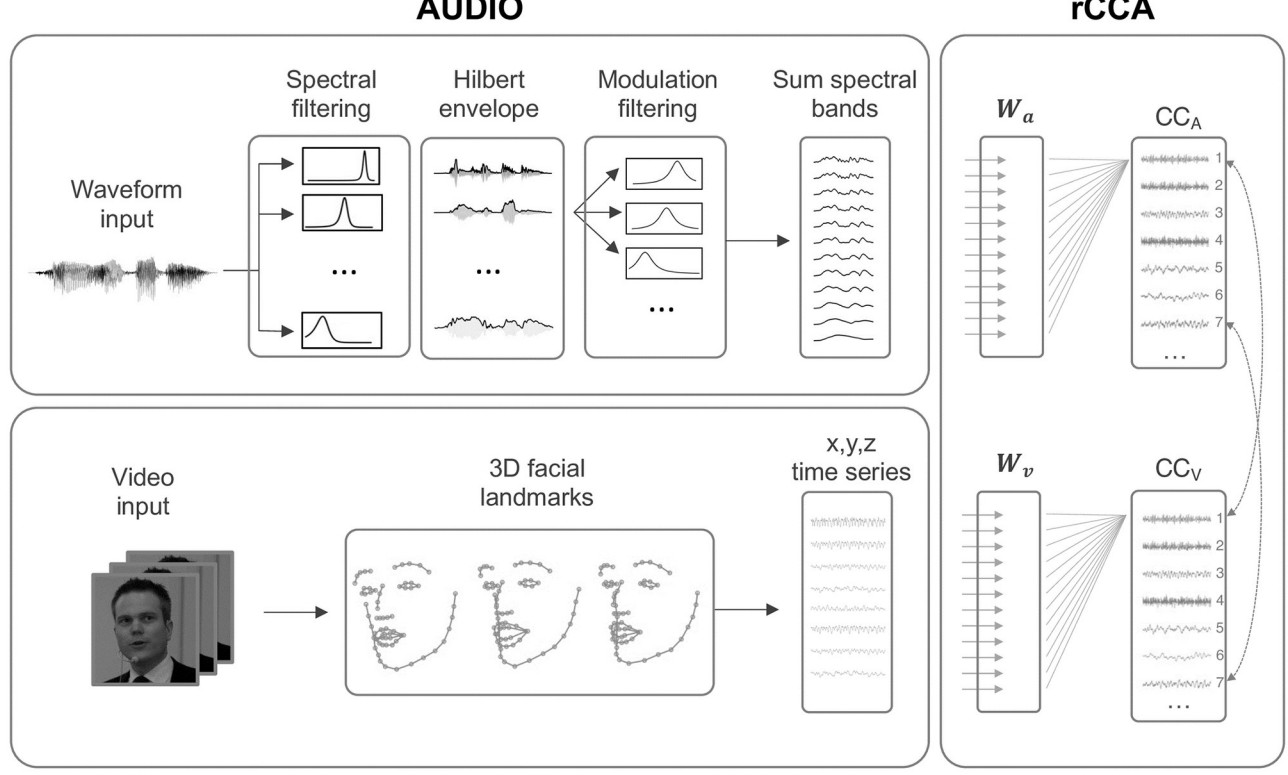

**Fig 1. Analysis procedure.** Regularized CCA (rCCA) combines speech envelope filter outputs and 3D landmarks of the speaker's face. Resulting pairs of canonical components (CCs) are linear combinations of envelope filter outputs for audio ($CC_A$) and facial landmarks for video ($CC_V$). Image source: commons.wikimedia.org/wiki/File:Dr_H._L._Saxi_18_April_2013.jpg.

**Visual feature extraction.** 3D-facial landmarks were extracted from the videos on a frame-by-frame basis. The landmarks were extracted using the deep learning-based face alignment network presented in [53]. The network first performs face identification in a given frame and then estimates the 3D position of 68 facial landmarks (see Fig 1). Each landmark is composed of an $x$, $y$, and $z$ coordinate, where the $x$ and $y$ coordinates correspond to the pixel location of a given landmark in the image frame, and the $z$ coordinate is the estimated depth location of the landmark.

The landmark time series were first low-pass filtered at 8 Hz to remove jitter in the frame-to-frame estimation and shifted by one sample. Energy above this range is unlikely to stem from speaker motion that can be detected at the video sampling rate of 25 Hz [20]. Finally, for each video the landmarks were normalized to have zero mean and unit variance in each of the three spatial ($x$, $y$, $z$) dimensions.

## Canonical correlation analysis

Given two multidimensional datasets, CCA finds linear transforms that project each dataset to a shared space where they are maximally correlated. Let $\mathbf{X_A} \in \mathbb{R}^{T \times J_A}$ and $\mathbf{X_V} \in \mathbb{R}^{T \times J_V}$ be two zero-mean datasets, where $T$ denotes time, and $J_A$ and $J_V$ are the number of features in the two datasets. CCA estimates pairs of vectors $\mathbf{w_{Aj}}$ and $\mathbf{w_{Vj}}$ such that the projections of the centered data $\mathbf{X_A}\,\mathbf{w_{Aj}}$ and $\mathbf{X_V}\,\mathbf{w_{Vj}}$ are maximally correlated:

$$
\begin{aligned}
\rho &= \max \frac{(\mathbf{X_A}\mathbf{w_{Aj}})^\top (\mathbf{X_V}\mathbf{w_{Vj}})}{\|\mathbf{X_A}\mathbf{w_{Aj}}\| \, \| \mathbf{X_V}\mathbf{w_{Vj}}\|} \\
&= \max \frac{\mathbf{w_{Aj}}^\top \Sigma_{AV} \mathbf{w_{Vj}}}{\sqrt{\|\mathbf{w_{Aj}}^\top \Sigma_A \mathbf{w_{Vj}}\| \, \| \mathbf{w_{Vj}}^\top \Sigma_V \mathbf{w_{Vj}}\|}}.
\end{aligned}
\tag{1a}
$$

where $\Sigma_A = \mathbf{X_A}^\top \mathbf{X_A}$ and $\Sigma_V = \mathbf{X_V}^\top \mathbf{X_V}$ are the (unnormalized) covariance matrices and $\Sigma_{AV} = \mathbf{X_A}^\top \mathbf{X_V}$ is the cross-covariance. Projections of the data $\mathbf{X_A}\,\mathbf{w_{Aj}}$ and $\mathbf{X_V}\,\mathbf{w_{Vj}}$ are denoted the canonical variates or *canonical components* (CCs). The first CC pair is the linear transformation of the datasets yielding the highest correlation. The next $J$ pairs of canonical components have the highest correlation while being uncorrelated with the preceding component. The components are thus ordered with respect to the size of correlation. In matrix notation, CCA yields two weight matrices $\mathbf{W_A} \in \mathbb{R}^{J_A \times J_0}$ and $\mathbf{W_V} \in \mathbb{R}^{J_V \times J_0}$, where $J_0 \leq \min\{J_A, J_V\}$, such that pairs of columns of $\mathbf{X_A}\mathbf{W_A}$ and $\mathbf{X_V}\mathbf{W_V}$ (the CCs) are maximally correlated. The CCs can be found iteratively, but since scaling of the canonical weights does not change the correlations, we can add the constraints that $\mathbf{w_{Aj}}^\top \Sigma_A \mathbf{w_{Aj}} = 1$ and $\mathbf{w_{Vj}}^\top \Sigma_V \mathbf{w_{Vj}} = 1$, and hence reformulate Eq (1a) as a Lagrangian that can be solved as a generalized eigenvalue problem.

CCA can be regularized to avoid overfitting. An L2 regularization term can be incorporated into the objective function in Eq (1a):

$$
\rho = \max \frac{\mathbf{w_{Aj}}^\top \Sigma_{AV} \mathbf{w_{Vj}}}{\sqrt{(\mathbf{w_{Aj}}^\top \Sigma_A \mathbf{w_{Aj}} + \lambda_A \|\mathbf{w_{Aj}}\|^2)(\mathbf{V_{Vj}}^\top \Sigma_V \mathbf{w_{Vj}} + \lambda_V \|\mathbf{w_{Vj}}\|^2)}}
\tag{2}
$$

Note that by adding regularization we effectively relax the constraint that $\mathbf{w_j}^\top \Sigma \mathbf{w_j} = 1$.

## AV modulation transfer functions

Here we use CCA to simultaneously learn a set of temporal modulation filters and spatial decompositions of the facial landmarks. The CCA analysis pipeline is illustrated in Fig 1. $\mathbf{X_A}$ is

the data matrix of $J_A$ (25) filtered subband audio envelopes, and $\mathbf{X_V}$ is the data matrix of visual features of size $T \times J_V$, where $J_V$ is the total number of facial landmarks ($3{*}68$). We assume that linear combinations of audio and video features are correlated by virtue of both being generated by the same speech production source. Specifically, let $\mathbf{X_A} = \mathbf{SA_A} + \epsilon_A$ and $\mathbf{X_V} = \mathbf{SA_V} + \epsilon_V$ be a forward 'generative' model, where a set of speech production sources $\mathbf{S} \in \mathbb{R}^{T \times J_0}$ generate both envelope fluctuations in the audio signal $\mathbf{X_A}$ and spatial motion in the face points $\mathbf{X_V}$. Columns of $\mathbf{A}$ are filters that map between the speech source and the observed audio envelopes and video landmarks, i.e. spectral filters in $\mathbf{A_A}$ and spatial filters in $\mathbf{A_V}$. CCA can now produce two transform matrices $\mathbf{W_A}$ and $\mathbf{W_V}$ that instead map 'backwards' from the observed features to estimate a set of latent sources (the CCs), i.e. $\hat{\mathbf{S}} = \mathbf{X_A W_A}$ and $\hat{\mathbf{S}} = \mathbf{X_V W_V}$. However, the CCA weights cannot be directly interpreted as the filter parameters $\mathbf{A}$ in the corresponding forward model [54]. The size of the CCA weights reflects both a weighting of those AV features that are correlated (particular combinations of envelope subbands and spatial landmarks), but also a suppression of 'noise', i.e. envelope fluctuations or visual motion that are not related to the shared speech source. However, the parameters of the corresponding forward model can be estimated as $\hat{\mathbf{A}} = \Sigma\mathbf{W}$ [54], also referred to as the canonical loadings. Unlike the CCA weights, the columns of the $\Sigma\mathbf{W}$ matrix indicate the correlation between CCs and the input features, i.e. the strength of the latent speech source in each of the observed features.

For the audio envelope features, each CC learned by CCA represents a weighted sum of the envelope subbands from the outputs of the modulation filterbank. Due to the distributivity of convolution, an additive signal summed at the output of an N-channel parallel filterbank with impulse responses $h_1, h_2, .., h_N$ is equivalent to filtering the input signal with a filter given by the sum of impulse responses $h_1 + h_2+, ..+ h_N$. The effective modulation transfer function learned by CCA is therefore given by the weighted sum of the impulse responses of the modulation filterbank. If $\mathbf{H} \in \mathbb{R}^{F \times J_A}$ is the set of transfer functions for the modulation filterbank with $J_A$ channels and F frequencies, the effective MTFs learned by CCA is thus given by $\mathbf{H}\Sigma_A \mathbf{W_A}$.

The MTFs can be visualized by inspecting the CCs, i.e. the output of the learned filters. In the results below, we plot the average spectra of the component time series $\mathbf{X_A} \Sigma_A \mathbf{W_A}$ computed for each video and each CC. This takes the average modulation energy across speakers in the dataset into account, i.e. it shows the effective outputs of the filtering learned by CCA.

On the visual side, CCA decomposes the facial landmarks into spatial groups with correlated motion. The landmarks corresponding to each CC can similarly be visualized in the face by the canonical loadings, i.e. the CCA weights for each landmark scaled by the sample covariance $\Sigma_V \mathbf{W_V}$.

## Optimization scheme

To identify statistically significant AV correlations that generalize across speakers, we trained the rCCA model using a cross-validation scheme. The dataset was first split into a test set and a training set consisting of 10% and 90% of the data, respectively. Cross-validation was then performed on the training set by further splitting the training data into five folds. Importantly, no speakers appeared in more than one data split, both for the test and training sets and for the individual cross-validation folds. This implies that the model was optimized to predict AV correlations across speakers. The rCCA was trained using a match-mismatch scheme [55]. During cross-validation, rCCA models were trained on correctly matching video and audio data on four of the five folds, and correlations for each rCCA component were computed on the held-out validation fold. Correlations for each component were then computed on 1000 mismatching segments of audio and video to generate an empirical null-distribution. The

difference between the median correlation obtained from the mismatching data and the correlation for the matching data defined the objective function that was used to optimize the two regularization parameters. Only matching components exceeding the 95th percentile of the null-distribution were considered.

For optimization, Bayesian Optimization via Gaussian Processes was used. The optimization scheme was implemented using `scikit-opimize.gp_minimize` [56]. The search space for both regularization parameters, RegA and RegV, were chosen to be between $[10^{-5}, 10^{0}]$. The `scikit-opimize.gp_minimize` algorithm was initialized with a random search for the two regularization parameters, which were drawn from a log-uniform distribution with upper and lower bounds defined by the search space. After evaluating the five random searches, the algorithm approximated the next five regularization parameters with a Gaussian process estimator using a Matern kernel. The `gp_hedge` acquisition function was used, which chooses probabilistically among the three acquisition functions: lower confidence bound, negative expected improvement, and the negative probability of improvement, at each iteration. This process was repeated for each of the five validation folds, and the regularization parameters yielding the highest difference in correlations across the five-folds were used to train a final rCCA model on the entire training set. The significant rCCA components of this final rCCA model were determined on the independent test set. Significant components were defined as those exceeding the 95th percentile of the null-distribution obtained with mismatching audio and video. We do not report CCs with an average correlation on the test set below 1%, even if they are significant.

## Results

We used CCA to relate speech envelope information and facial motion across a large number of speakers ($\sim 4000$). Specifically, CCA learns envelope filterings that correlate with visual motion in groups of facial landmarks (see Fig 1). Fig 2 shows statistically significant canonical components (CCs) for the main analysis on the LRS dataset. Importantly, the significance of the CCs was determined by whether they generalize across talkers. The left panels show the outputs of the envelope filters learned by CCA for each CC. The right panels show the corresponding contribution of facial landmarks visualized by the 2D projection of the landmark CCA loadings. The color bars indicate the relative contribution of the *x,y,z*-directions. A dynamic visualization of the facial CCs for an example speaker can be seen in S1 Video. This example is not a facial animation but a dynamic plot of the visual CCs back-projected to the input landmark space to aid interpretation of how the CCA decomposes face and head movements during speech.

The first canonical component CC1 represents the largest correlation between the AV features. As can be seen in Fig 2, CC1 extracts motion of the lower lip and jaw, mainly in the vertical direction, which is correlated with speech modulations at rates peaking around 3–4 Hz. CC4 complements CC1 by extracting envelope information in a similar envelope frequency range with a peak around 4 Hz, but correlated with vertical movement of the upper lip and upper parts of the head. Together, CC1 and CC4 represent a modulation transfer function for the envelope that aligns with the average modulation spectrum for natural speech, with a peak around 3–4 Hz [7, 15]. Our analysis indicates that this 4 Hz peak is statistically correlated with two main sources of visual face motion centered at the lower and the upper parts of the mouth. The first (CC1) relates to mandibular motion that can be performed relatively independently of other head movements. The second (CC4) relates to maxillary movements that are naturally coupled with pitch axis rotations of the head relative to the mandible. These two components

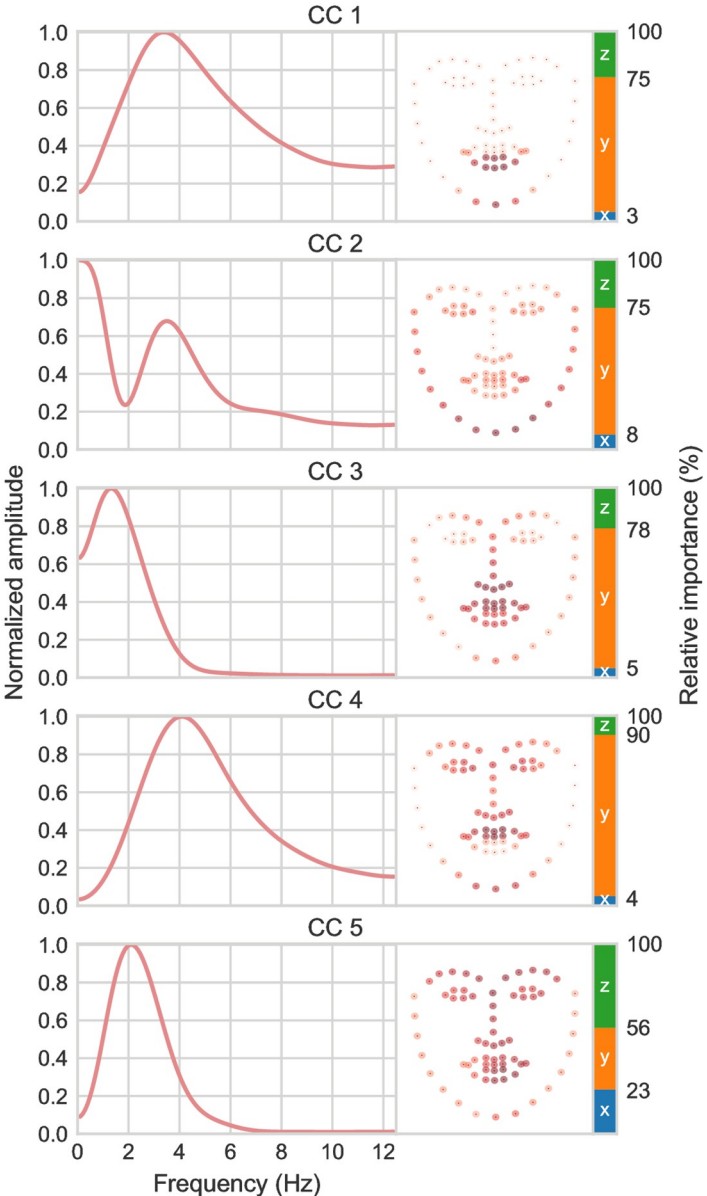

**Fig 2. CCA results for the LRS3 dataset.** *Left*: CCA-derived temporal modulation filters for the first 5 significant canonical components (CCs). *Right*: corresponding facial landmark loadings. Darker red indicates higher weights. The 3D landmarks are shown in 2D projection, and the colorbar indicates the relative contribution of the *x* (blue), *y* (orange), and *z* (green) directions.

thus appear to capture two main kinematic dimensions of mouth open-close cycles during speech production.

The envelope frequencies associated with mouth openings (CCs 1 and 4) are relatively broadly distributed around 4 Hz. This may partly reflect variation in e.g. speaking rate across talkers [13, 14]. To investigate this, we computed the spectral peaks of the envelope CCs separately for each video (and thus for each speaker) in the dataset (see S1 Fig). The distribution indeed matches the shapes of the filters learned by CCA.

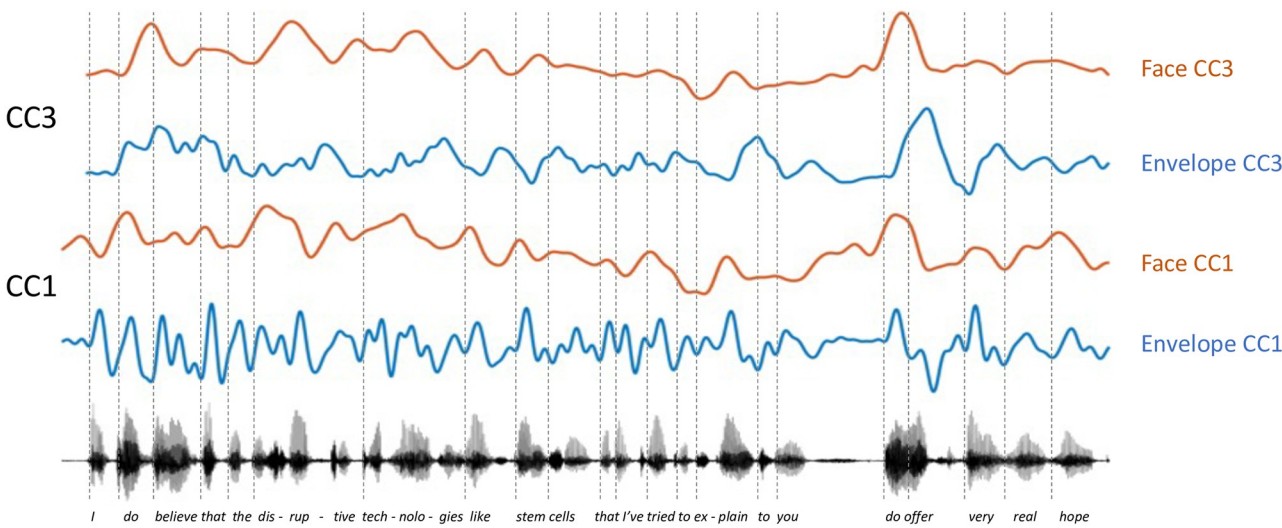

**Fig 3. CC1 and CC3 for an example speaker.** The CC time series for the speech envelope are shown in blue, and the CCs for the facial landmarks are shown in orange. Vertical lines indicate word onsets. CC1 represents speech envelope fluctuations corresponding to the onset of individual syllables, while CC3 tracks slower variations corresponding to words or phrases.

Whereas CC1 and CC4 capture mouth openings correlated with envelope rates distributed around 3–4 Hz, CCs 2, 3 and 5 capture slower modulations around 1–2 Hz correlated with more global head and face movements. CC3 specifically extracts pitch axis rotations of the head, whereas CC5 relates to rigid head movements in all spatial directions. The spatial decomposition learned by CCA thus isolates rigid 3D head rotations by a single component (CC5) while removing $x$ and $z$ rotations from the remaining components. While CCs 3 and 5 capture head rotations, loadings on oral landmarks are also high, in particular for CC3. This indicates that head and mouth movements are mutually correlated and together correlated with slower speech envelope information. This occurs, for instance, when head nods are synchronized with certain mouth openings to produce accents on important words, thereby yielding envelope fluctuations at a slower rate. CC2 appears to combine envelope information at the two rates in one component.

Together, the visual face and head appear to carry speech envelope information at two distinct timescales during natural speech. Envelope fluctuations peaking around 3–4 Hz are specifically associated with mouth openings (CCs 1 and 4), while slower 1–2 Hz modulations are correlated with coordinated motion across the face and head (CCs 2, 3, 5). For illustration, CC1 and CC3 for an example talker are plotted in Fig 3. As can be seen, modulations around 3–4 Hz captured by CC1 track speech at the level of syllable onsets, while the slower 1–2 Hz modulations of CC3 capture variations at the level of phrases. Local time shifts between face and envelope CCs can occur as can be seen when inspecting CC loadings for individual speakers. For instance, in the example shown by CC3 in Fig 3, a vertical head rotation used to emphasize the final statement ('*do offer*') precedes the acoustic modulation associated with the produced stress.

Because of the data-driven nature of the analysis, it is important to determine the consistency of the learned AV components. To investigate reliability, we split the dataset into two equal halves and performed the same analysis separately on each split. None of the speakers overlapped between the two halves. The results of the split-half analysis are shown in S2 Fig. As can be seen, the CCA-derived envelope filters and corresponding face loadings are highly similar in the two separate analyses. This indicates that the observed temporal regularities are

stable when considering AV speech statistics across many speakers. S3 Fig also illustrates this point by showing MTFs for CCA solutions computed with a varying number of speakers. With increasing amounts of data, the bandpass filter shapes become increasingly stable, in particular for higher components.

### Analysis of the GRID dataset

As a supplemental analysis, we performed the same rCCA analysis on the GRID speech database. Unlike the LRS videos of natural speech *in the wild*, the GRID corpus consists of videos of a smaller number of speakers (34) instructed to perform simple and syntactically identical monosyllabic sentences (such as 'put red at G9 now') [51]. Movements beyond those involved in sound production are thus minimized in this data. The GRID data is comprised of numerous videos from each speaker, whereby the total amount of data included in the GRID analysis was similar to the analysis of the LRS data.

The components learned for the GRID data are shown in Fig 4. Again, components that generalize significantly across speakers are shown. As can be seen, CCA again learns envelope filters distributed around 3–4 Hz. CCs 1, 2, and 5 again capture mouth openings and associated movements of the lower (CC1, CC2) and upper (CC5) parts of the face, highly similar to CCs 1 and 4 found for the LRS data. Unlike the LRS data, however, all components for the GRID data have envelope filter peaks in the 3–5 Hz range and relate more closely to orofacial motion. In addition to the upper and lower part of the mouth, regions around the two lip corners emerge as separate CCs (CC3, CC4, CC7). Slower envelope rates in the 1–2 Hz range related to head motion as in the LRS dataset are not apparent in the GRID data analysis. Instead, the GRID data highlights several details of oral motion.

## Discussion

In the current study, we present a CCA technique to learn speech envelope filterings that are correlated with visual face motion. Our analysis relates different rates of acoustic envelope variation to visual motion in different parts of the talking face. The main results for the LRS natural speech dataset indicated two primary temporal ranges of envelope fluctuations related to facial motion across speakers. The first is distributed around 3–4 Hz and relates to mouth openings. The second range of modulations peaks around 1–2 Hz and relates to more global face and head motion. Envelope information at both rates were correlated with landmarks distributed across the face, in agreement with the fact that natural speech involves highly coordinated motor activities. This also implies that many speech envelope cues may not only be available from mouth movements but can be retrieved from non-oral parts of the face and head. Importantly, the derived AV correlations were predictive across different speakers implying that these temporal cues are consistent in natural AV speech statistics.

### Bandpass envelope MTFs

Our analysis revealed modulation transfer functions with a bandpass character. A number of previous studies have investigated the relation between speech envelopes and facial movement, e.g. by correlating motion data with the low-passed Hilbert envelope of the audio waveform [5, 6, 18–20]. However, our analysis indicated that envelope information is correlated with visual face motion at specific temporal scales. This echoes the sensitivity of the auditory system to envelope information at different timescales [57]. In the auditory domain, bandpass-like modulation sensitivity has been modeled as a modulation filterbank, with filters acting as AM detectors at different rates [44, 58]. For instance, accurate prediction of speech intelligibility in fluctuating noise maskers has been argued to rely on the signal-to-noise ratio in the envelope

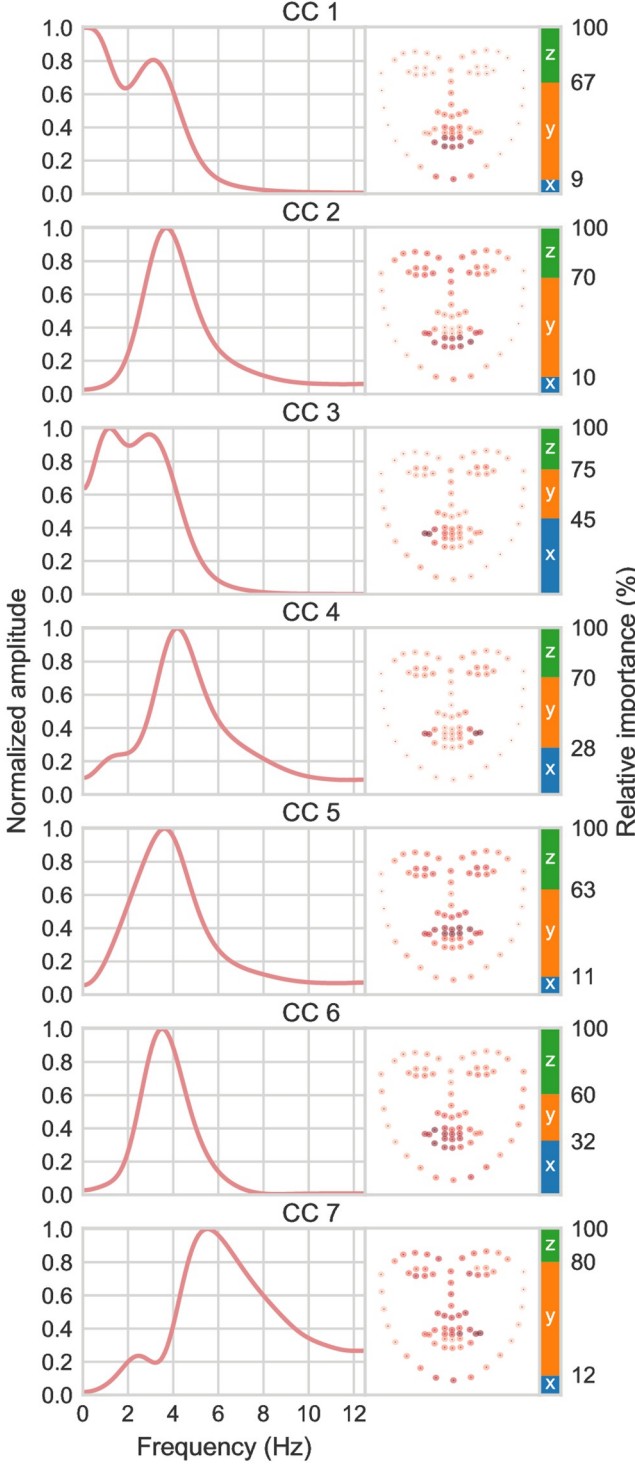

**Fig 4. CCA results for the GRID dataset.** CCA-derived envelope filters (*left*) and corresponding face loadings (*right*) for the GRID dataset. Unlike *in the wild* recordings of natural speech such as the LRS3, the GRID corpus is composed of simple, syntactically identical six-word sentences.

domain, i.e. after modulation-frequency selective filtering [59]. While sensitivity to higher modulation frequencies may be unique to audition, slower temporal cues may be processed in a multisensory fashion [6, 60]. Our analyses indicate that AV envelope cues are available at two distinct timescales below 10 Hz. These are not simply different low-passed versions of the broadband envelope, but bandpass modulation filters in the 1–2 Hz and 3–4 Hz ranges, respectively, that appear to capture distinct sources of correlation in natural AV speech.

## Two rates of AV regularity

These two rates of speech modulations correspond well to the rates at which syllables (3–4 Hz) and phrases or prosodic features (1–2 Hz) are produced in natural spoken language [61, 62]. The onsets of individual syllables are pronounced energy transitions in a speech signal, as reflected by the fact that the average modulation spectrum is dominated by energy around 3–4 Hz [16]. Acoustic cues for segmenting a continuous speech signal into phrases are less prominent in the envelope spectrum, where energy falls off below 3 Hz. However, when considering speech as an audio-visual signal (rather than a purely acoustic one) slower envelope rhythms in the 1–2 Hz range emerge as a distinct range of temporal regularity. AV correspondences at these two different timescales may thus provide cues for segmenting the continuous speech signals at the level of syllables and phrases.

This might also indicate a motor origin of temporal regularities at these two distinct timescales. Rhythmic head or limb movements performed during speech are typically slower than mouth movements involved in syllable production. Head nodding or hand gestures during speech have been reported to be synchronized with envelope or pitch variations below 2 Hz [29, 30, 33, 63], consistent with our analyses. Mouth open-close cycles during speech, on the other hand, matches the natural syllable production rate around 4 Hz [6, 12]. Different temporal regularities imposed by these oral and non-oral motor components may emerge in facial communication before language and persist in speech. It has been proposed that the use of faster mouth movements to produce acoustic modulations at the syllable rate may be a unique adaptation in humans [64]. MacNeilage (1998) proposed that the motor capacity for rhythmic orofacial control in speech may have evolved via slower ingestion-related mandibular cycles. Macaque monkeys can produce rhythmic vocalizations in the 3–4 Hz range (i.e. vocalizations with modulation spectra similar to speech) accompanied by a single facial movement trajectory, rather than by synchronized open-close cycles of the mouth [65]. Faster cyclic movements of the jaw, lips, and tongue in the 3–7 Hz range are used in non-vocal visuofacial communication (lipsmacking, teeth chattering) in non-human primates [36], and may have been adapted for vocal behavior in humans [65–67]. A parallel transition between two rates of vocal production can be observed in human speech development. In the first year of life, infants begin to produce rhythmic babblings (repeated consonant-vowel-like sequences like 'bababa') synchronized with mouth open-close cycles that are below 3 Hz [68] and coordinated with rhythmic limb movements [69–73]. From slower and more variable vocal rhythms in infancy, faster and more regular envelope-mouth synchronization above 4 Hz as in adult speech emerge gradually during development [8, 74].

Thus, slower vocalizations coordinated with limb movements can be viewed as a precursor to faster vocalizations synchronized with mouth openings at the syllable rate [71, 75]. However, speech modulations at the syllable rate do not necessarily replace slower modulations, but may be superimposed on them. Our analysis points to the co-existence of two unique sources of AV correlation, e.g. slower (1–2 Hz) rates of speech modulations synchronized with head and face movement co-exist with faster mouth-envelope synchronization.

The two distinct rates of AV correlation only emerged in our analyses when considering natural speech across many speakers. Analysis of the GRID data highlighted the well-known synchronized mouth-envelope modulations in the 4 Hz range [6]. However, the analysis across many speakers in the LRS dataset revealed the slower timescale to be a consistent source of AV correlation in natural speech. These differences between datasets suggest an interesting predisposition in AV speech studies. Controlled speech production, as in the GRID matrix sentences, strips away important gestural features that are prominent in natural speech. Speech can be produced with minimal gestural movement [76], but gestures consistently accompany natural speech [77]. Gestures occur even in conversations between blind people [78], suggesting a nonincidental association. Analysis of the GRID data confirmed the prominence of speech modulations distributed around 4 Hz [7, 17] correlated with mouth open-close cycles [6], but the analysis does not fully reflect the prominence of envelope information below 4 Hz in natural speech. It also does not fully capture the degree to which envelope information is consistently correlated with motion in many different parts of the face. Different data splits within each dataset yielded highly consistent CCA components (S2 Fig), indicating that differences between the two datasets stem from differences in the nature of the data. Different speech materials based on different speech tasks thus appear to implicitly zoom in on particular features of AV speech.

## AV decomposition of the speaking face

While slower modulations were not found in the analysis of the GRID data, the GRID data revealed a number of more detailed orofacial components. Decomposition of the face during speech has been pursued in previous work using PCA [18, 19, 37], ICA [79] or other matrix factorization algorithms [38, 40]. Lucero et al. (2008) identified independent kinematic components for the upper and lower parts of the mouth and the two mouth corners, that were also identified in our AV analysis of the GRID data [40]. In contrast to previous work, our CCA performs a joint dimensionality reduction in the visual and auditory domain to identify facial regions that are correlated with envelope information. The GRID analysis indicated that the different local kinematic regions of the mouth (upper lip, lower lip, left and right corners), also found in visual-only face decompositions [40], correlate with envelope information in the 3–4 Hz range. The independent kinematics of lip corners could potentially relate to grimacing unrelated to acoustic information (e.g. smiling), but this does not appear to be the case. Other spatially local components, such as the eyes or eyebrows that appear as independent components in visual-only decompositions of the face [40], were not identified as isolated components in our AV analysis. However, a number of components showed high loadings on landmarks around the eyes and upper parts of the face in combination with oral ones. This suggests that e.g. raising of the eyebrows at prosodic events [80] is consistently coupled with movement in other parts of the face. While non-oral facial parts, such as the eyebrows, may display independent kinematics [40], only movements that are coordinated across the face are consistently correlated with envelope information in our analysis. This high redundancy also implies that similar envelope information is available from many parts of the face.

## Neural sensitivity

We note that the two distinct modulation frequency regions emerging from our AV analysis align noticeably with the modulation sensitivity of human auditory cortex. Human auditory cortical activity is known to track envelope fluctuations at distinct rates below 10 Hz in speech or other natural stimuli [81]. Speech envelope tracking occurs specifically in the theta (4–8 Hz) and delta (1–3 Hz) frequency bands of the human electroencephalogram [82–85], and

synchronization of cortical activity in these bands have been proposed as a neural mechanism for parsing speech at the level of syllables and phrases [86]. Yet, the fact that these same modulation frequency regions emerge from AV signal statistics could suggest that temporal modulation tuning in auditory cortex is adapted to the statistics of natural AV stimuli. Auditory cortex is known to integrate correlated visual signals [87–89], and AV correlations at different timescales may have shaped band-pass modulation selectivity in auditory cortex, persisting with auditory-only inputs. Rather than a language-specific mechanism for tracking syllables and phrases, cortical envelope tracking specifically in the delta and theta ranges may thus reflect a cortical envelope tuning adapted to temporal regularities that are ultimately determined by auditory-motor constraints.

## Perceptual relevance

Our analyses suggest the availability of temporal cues at distinct rates from different parts of the face, but not how these are used in perception. It is well known that viewing a talker's mouth aids auditory speech perception [2, 90, 91]. Degrading visual temporal cues, e.g. by reducing the frame rate in videos of the speaker's face, reduces the AV perception benefit [92, 93]. Non-oral facial movements also contribute to AV perception as seen by the fact that AV perception benefits occur when the mouth is visually occluded [94]. Seeing head motion can improve speech intelligibility [29] and has been argued to provide prosodic speech cues [24–28, 77]. This is consistent with our analyses indicating an association between slower envelope information and head movement. While envelope information distributed around 3–4 Hz was closely related to mouth openings, these components were also correlated with non-oral facial landmarks. This also implies that envelope information at both timescales is available when only seeing parts of the face. Temporal modulations at these rates are particularly important for speech intelligibility [45, 95], making coordinated movements across the face a useful perceptual cue. Being distributed across the face, temporal modulations are likely not perceived via the motion of individual speech articulators, but as motion patterns of coordinated facial components. Johnston et al. (2021) recently reported that subjects were highly sensitive to the degree of synchronicity between mouth and eyebrow motion, suggesting that coordinated motion across the face facilitates perceptual binding [96].

## Modelling AV speech across speakers

In contrast to much earlier work, our analysis takes a between-speaker approach to AV speech. Our CCA analysis scheme was designed to extract AV statistics that are predictive across many speakers. Much finer details of face-speech correlation can be observed at the individual level, but speaker-specific analyses do not reveal which AV patterns generalize across talkers. Ginosar et al. (2019) recently proposed a deep neural network model that predicts hand gestures of an individual speaker from speech audio of that speaker [97]. Models were trained on large amounts of data from few speakers in order to synthesize the gestural styles of the individual speakers convincingly. In contrast, we focused our analysis on little data from a large number of speakers in order to identify AV speech-face correlations that are predictive across speakers. The person-specific approach of Ginosar et al. (2019) and others was motivated by the argument that speech gesture is essentially idiosyncratic [77], and that different speakers use 'different styles of motion' [97]. While speaker-specific models may indeed capture most variance in speech gesture data, our between-speaker approach demonstrates that important aspects of AV gesture generalize across talkers. It is perhaps unsurprising that mouth movements directly associated with speech production generalize across talkers, but also AV components related to more global gestural head movements appear to generalize. Although gestures like hand or

head movement may have acoustic consequences [30], speech can be produced with limited gestural movement [26, 76], and their consistency across speakers must be established empirically.

## Applications

Previous work has used CCA for audiovisual applications, such as speech separation [98], audiovisual synchronization [99, 100], or facial animation [101]. In such applications, feature extraction is typically performed to optimize the performance of the particular application. Here, we focused on learning generalizable features that are informative about AV speech, but relevant applications can also be highlighted. Our approach regularizes the CCA across speakers to identify features that are consistently correlated across talkers, making the approach attractive for AV speaker identification. For instance, our approach can be used to identify which of N separated audio sources (e.g. from an acoustic source separation system) belongs to which talking face in multi-talker video data (see S4 Fig). CCA is a linear technique and the feature transforms are fast to compute, making them appealing for real-time applications.

## Limitations

Some limitations in the current approach must also be highlighted. First, our analysis does not account explicitly for time lags between the audio and video. The degree to which audio might lag visual speech is debated [6, 102]. Speech gestures such as head nods do not have to occur simultaneously with the speech [76], and time lags may vary between speakers [27]. This individual variation is explicitly ignored in our between-speaker approach. CCA can readily be extended to account for time-lags [41, 103] (see also [104]). However, a narrowly spaced envelope filterbank covering low modulation frequencies is likely to be able to absorb time shifts between the signals [41], at least within the temporal range normally considered to be relevant for AV integration [105].

Speech datasets like the LRS3 enable large-scale studies of AV statistics across speakers, but the nature of the data also limits such investigations. The differences between our analysis of natural speech in the LRS dataset and the GRID dataset illustrate the fact that differences in the data influence the results. While the recordings of TED talks in the LRS dataset can be considered as representing natural speech, the data still represent largely scripted monologues. Most natural speech occurs in the form of dialogues or conversations involving spontaneous turn-taking. Speech rhythms during turn-taking may be adapted to the temporal structure of turn-taking behavior [25, 106–108], which may not be captured when analyzing video of monologues. Unfortunately, large video speech datasets involving natural communication are currently sparse.

Our analyses focus specifically on quantifying transfer functions for the speech envelope. Identifying envelope-face correlations with natural speech across many speakers is likely to favor visual and acoustic sources that have large variance. Temporal envelope features are also limited to low frequencies by the video sampling rate. While it is widely accepted that slow modulations below 10 Hz are important for speech perception [5, 93], many other more finegrained features are clearly essential in AV speech perception. In particular, our analyses ignore spectral information. Features such as line spectral pairs or Mel-Frequency Cepstral Coefficients (MFCCs) that are sensitive to local phonetic contrasts have been shown to correlate with face movements [19, 21, 109] and could be explored.

It should also be noted that 3D facial landmarks can pick up physiological motion such as heart rate or breathing patterns [110, 111] which occur at similar low frequencies to those revealed in our AV analysis. If these are synchronized with speech envelopes [112] they may

be picked up by CCA. However, motion of the face and head during speaking are typically larger in amplitude compared to motion caused by such physiological parameters and physiological parameters are therefore not likely to contribute significantly to the higher components investigated here.

Importantly, CCA is a linear technique and our approach only considers linear relations between visual and acoustic features. The relation between visible articulators and the produced speech signal is non-linear in important aspects [19, 20, 48, 109], and a linear model is therefore principally limited in capturing these. Yehia et al. (2002) found that a non-linear neural network outperformed a linear model in predicting head motion from acoustic features [18, 20]. Nonlinearities may, in principle, be accounted for by appropriately transforming the acoustic and visual features. However, here, the main goal was to learn these feature transformations from the AV speech data. The availability of extensive speech datasets and improved techniques for facial landmark estimation may enable data-hungry non-linear models to learn feature transformations from more simple input features. However, this arguably involves a trade-off between model accuracy and interpretability. In our approach, CCA learns a linear combination of linear envelope filters, which is itself an envelope filter. This implies that the components can be investigated directly in the envelope domain, i.e. we can directly investigate which envelope frequencies relate to motion in different parts of the face. The fact that results can be linearly related back to the input space arguably facilitates interpretation.

## Supporting information

**S1 Video. Video of reconstructed face CCs for an example speaker.** Backprojection of the first 5 facial canonical components (CCs) for an example speaker in the LRS3 dataset (ID 00j9bKdiOjk). *Left*: original estimated 3D facial landmarks. *Right*: reconstructed facial landmarks for CCs 1–5. The reconstructed landmarks illustrate the kinematic dimensions of facial motion captured by the individual CCs.
(MP4)

**S1 Fig. Distribution of spectral peaks on envelope CCs across videos in the LRS3 (left) and GRID (right) datasets.** The variation in spectral peaks across videos aligns with the shape of the CCA-derived modulation filters (Figs 2 and 4).
(TIF)

**S2 Fig. Split-half reliability.** The same CCA analysis was performed on two independent halves of the LRS3 dataset ($\sim$ 1950 different speakers in each split). Envelope filters (left panels) and spatial decompositions of the visual face (right panels) learned via CCA were highly similar between the two data splits.
(TIF)

**S3 Fig. MTFs for varying number of speakers.** MTFs were computed for different amounts of speakers by subsampling the data. For different numbers of speakers, nine different CCA solutions were computed while keeping regularisation parameters fixed. As can be seen, a higher number of speakers lead to more convergent solutions. We note that CCs 4 and 5 may switch place in different subsamples.
(TIF)

**S4 Fig. Speaker identification.** The AV CCA model enables fast speaker identification. Here, the CCA model is used to identify which of 2 (solid lines) or 3 (dashed lines) different audio segments correspond to 2 or 3 video segments. The AV pair with the highest correlation on CC1 is chosen as the matching pair. Only videos not used for training the CCA model were

used for speaker identification. Identification performance is shown as a function of AV segment duration for the LRS3 (blue) and GRID (orange) data. Shaded regions show ± SEM. (TIF)

**S5 Fig. Test correlations.** Test correlation values for the LRS3 (left) and GRID data (right). Boxes show null distributions derived by training CCA models on mismatching AV data. (TIF)

## Acknowledgments

We would like to thank Søren Fuglsang for helpful discussion during preparation of this manuscript.

## Author Contributions

**Conceptualization:** Nicolai F. Pedersen, Torsten Dau, Lars Kai Hansen, Jens Hjortkjær.

**Data curation:** Nicolai F. Pedersen, Jens Hjortkjær.

**Formal analysis:** Nicolai F. Pedersen, Jens Hjortkjær.

**Funding acquisition:** Torsten Dau, Jens Hjortkjær.

**Investigation:** Nicolai F. Pedersen, Jens Hjortkjær.

**Methodology:** Nicolai F. Pedersen, Lars Kai Hansen, Jens Hjortkjær.

**Project administration:** Jens Hjortkjær.

**Software:** Nicolai F. Pedersen.

**Supervision:** Torsten Dau, Lars Kai Hansen, Jens Hjortkjær.

**Validation:** Jens Hjortkjær.

**Visualization:** Nicolai F. Pedersen, Jens Hjortkjær.

**Writing – original draft:** Jens Hjortkjær.

**Writing – review & editing:** Nicolai F. Pedersen, Torsten Dau, Lars Kai Hansen, Jens Hjortkjær.

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
