## [Decision Letter · Decision Letter 0]

28 Mar 2022

Dear Dr. Hjortkjær,

Thank you very much for submitting your manuscript "Modulation transfer functions for audiovisual speech" for consideration at PLOS Computational Biology. As with all papers reviewed by the journal, your manuscript was reviewed by members of the editorial board and by several independent reviewers. The reviewers appreciated the attention to an important topic. Based on the reviews, we are likely to accept this manuscript for publication, providing that you modify the manuscript according to the review recommendations.

Sincerely,

Frédéric E. Theunissen

Associate Editor

PLOS Computational Biology

Samuel Gershman

Deputy Editor

PLOS Computational Biology

[LINK]

Dear authors,

Please respond carefully to the comments of reviewer1.

Best,

Frederic Theunissen

Reviewer's Responses to Questions

**Comments to the Authors:**

Reviewer #1: This impressive manuscript presents results from the use of regularized canonical correlation analysis on speech acoustics and video to learn temporal envelope filters that correlate with motion on different parts of the face. In both machine and human recognition of speech integration of audio and visual information from talkers enhances performance. This research provides an innovative and rigorous methodology to address part of this problem: What is the linear correspondence between the two signals?

The strengths of the work include that it analyzes a large AV data set as well as a more controlled smaller data set, a broad literature review considering the many issues (though a few key references noted below should be considered), and rigorous methods that consider the joint reduction in dimensionality for auditory and video signals to compute the correlated regions and timescales.

References to consider:

On the relationship between face movements, tongue movements, and speech acoustics

J Jiang, A Alwan, PA Keating, ET Auer, LE Bernstein - EURASIP Journal on Advances in Signal Processing, 2002

Quantifying time-varying coordination of multimodal speech signals using correlation map analysis

A Vilela Barbosa, RM Déchaine, E Vatikiotis-Bateson… - The Journal of the Acoustical Society of America, 2012

Grant, K. W., and Seitz, P. F. (2000). “The use of visible speech cues for improving auditory detection of spoken sentences,” J. Acoust. Soc. Am. 108(3), 1197–1208.

Yehia, H., Rubin, P., and Vatikiotis-Bateson, E. (1998). ‘‘Quantitative association

of vocal-tract and facial behavior,’’ Speech Commun. 26, 23–43.

The weaknesses of the manuscript include: A) In spite of the large data set and improved methods, the results are not really surprising. As the omitted references suggest, there are visual correlations between the acoustic envelope and parts of the moving head and face. The partition shown in this paper is nice but is partially anticipated by other studies that show additional correlation features. However, demonstrating something that is not surprising is still worth doing. B) The authors discuss a number of limitations but one that is inherent to all approaches like this is that components are dominated by larger variances sources. However, information that is perceptually salient can be quite small and statistically infrequent, but that information can make all the difference. For example, good lipreaders take advantage of high spatial frequency visual information in some contexts even though low spatial frequency aspects of the signal are generally fine. C) It seems that the correlations are computed on segments, thus with stationary AV temporal alignment. As the authors suggest temporal synchrony is a complex problem but perhaps more complex than they suggest. There are differences in physical transmission of speech sight and sound as well as differences in neural transmission and processing in the two modalities. But, also the two speech signals differ greatly in character. Acoustic speech can be segmented with more or less accuracy because it is punctuated by syllabic nuclei and silences. Visual speech is far more continuous and segmenting it is more similar to the problem of visual event perception (e.g., Zacks, 2020). In the past, this problem has been hacked by omitting the signals associated with silences (e.g., Yehia). This just avoids the problem. D) The temporal signal compression to match video and acoustic signals predetermines that only low frequency envelope filters will be observed. Granted that there can’t be much spectral power in the movements that is perceptually important (see de Paula et al.).

H. de Paula, H.C. Yehia, D. Shiller, G. Jozan, K.G. Munhall, E.Vatikiotis-Bateson

Linking production and perception through spatial and temporal filtering of visible speech information

Proceedings of the 6th International Seminar on Speech Production, Sydney, Australia, 7–10 December 2003 (2003), pp. 37-42.

But there is information in the acoustics above this low frequency filter level that can be predictive of movement and vice versa (e.g., in a Bayesian manner). In other words, correlation of temporally matched signals is not the only kind of binding.

Some additional comments:

Line 250… This is a possible null distribution but of all of the control distributions that could be computed this is certainly the weakest. Given the differences in speaking style and rate between talkers (as the authors show), this will produce a very uncorrelated distribution. But, if you wanted to control for differences in speaking rate and emphatic variance in speech, you could use utterances from the same talker or the same utterance played in reverse. I am not suggesting that this be done but be aware that this is a very null null. Perhaps the null distribution and the data correlations could be shown graphically in an appendix. This would make me understand the sentence on line 254-255. By the way, how many matching components were discarded as a result of that criterion.

Line 277 – correlations above 1%... What does that mean? And what percentage of the variance does correlation account for overall?

S3 Fig. Thus, the shape and peak of the group filters are determined by averaging smear. Note that the individual filters shown here are also averages for individuals’ speech sample and people change rate dynamically even within a phrase.

Line 359-361. I am happy to see this demonstration about the GRID type tasks being different from more natural speech. I presume that the lack of envelope rates in the 1-2 Hz range for the GRID tasks has something to do with them having no variation in prosody. The TED talks are more natural but are heavily scripted with edited retakes.

Lines 392-394. I am surprised that the envelope modulations in the two ranges were mutually uncorrelated. They should be related as stress and emphasis signalled by head motions should correlated with mouth aperture size. The different event rate and information span must wash that out.

Lines 376-379. This supports my view that analyses such as this are biased towards uncovering large movement variance as the comment on heart rate and breathing indicates.

In summary, this is a well carried out and thoughtful piece of work. It has sophisticated methodology and a number of good controls have been carried out to test alternative explanations. I feel that this manuscript will of interest to researchers in speech recognition and those who study the behavioral and neural concomitants of multisensory speech.

Reviewer #2: Review of Modulation transfer functions for audiovisual speech

In this excellent study, Pedersen et al. revisit the question of the link between auditory and visual cues in Speech. This is a well-studied and important field that has had extensive amount of work. The known result is that visual and auditory cues are roughly correlated in 3-8 Hz frequency bands and that head movement is often in the 1-2 Hz region and is important for speech perception. However, what is missing is a precise quantification of the precise rates at which envelope information is synchronized with motion in different parts of the face.

To address this question, the authors combined a novel approach that first used a deep learning approach to identify face landmarks, and a gammatone filter bank to filter the auditory signals. They then used regularized canonical correlation analysis (rCCA) to learn speech envelope filters whose outputs correlate with motion in different parts of the speakers face. In general, I found the paper to be well written and has some interesting conclusions, albeit consistent with prior work. I had a few comments which will hopefully improve the manuscript.

Statistical significance of components: Could the authors add what correlation is explained by each of the CCs? The authors say they report CCs where correlation was above 1%. That seems quite low.

Second, I feel the authors are not doing justice to their incredible 4000 speaker dataset by only showing the average correlation. I feel S1 Fig should perhaps be a side panel to the figure 1 and figure 1 should have errorbars around it. Reporting the correlation coefficient next to the canonical components would help the reader. The normalization also makes it hard to assess if some components are larger or smaller. I would say the authors should also include statistics and errorbars for the GRID corpus dataset.

**Have the authors made all data and (if applicable) computational code underlying the findings in their manuscript fully available?**

Reviewer #1: None

Reviewer #2: Yes

PLOS authors have the option to publish the peer review history of their article (what does this mean?). If published, this will include your full peer review and any attached files.

Reviewer #1: No

Reviewer #2: No

Figure Files:

Data Requirements:

Reproducibility:

References:

---

## [Editor Report · Decision Letter 1]

1 Jun 2022

Dear Dr. Hjortkjær,

We are pleased to inform you that your manuscript 'Modulation transfer functions for audiovisual speech' has been provisionally accepted for publication in PLOS Computational Biology.

Best regards,

Frédéric E. Theunissen

Associate Editor

PLOS Computational Biology

Samuel Gershman

Deputy Editor

PLOS Computational Biology

Thank you for your thorough response to the reviewers. Nice contribution.

F

---

## [Editor Report · Acceptance letter]

4 Jul 2022

PCOMPBIOL-D-22-00122R1 

Modulation transfer functions for audiovisual speech

Dear Dr Hjortkjær,

I am pleased to inform you that your manuscript has been formally accepted for publication in PLOS Computational Biology. Your manuscript is now with our production department and you will be notified of the publication date in due course.

With kind regards,

Marianna Bach
